# Ductal Carcinoma In Situ Underestimation of Microcalcifications Only by Stereotactic Vacuum-Assisted Breast Biopsy: A New Predictor of Specimens without Microcalcifications

**DOI:** 10.3390/jcm9092999

**Published:** 2020-09-17

**Authors:** Yun-Chung Cheung, Shin-Cheh Chen, Shir-Hwa Ueng, Chi-Chang Yu

**Affiliations:** 1Department of Diagnostic Radiology, Chang Gung Memorial Hospital, Medical College of Chang Gung University, Taoyuan 333, Taiwan; alex2143@cgmh.org.tw; 2Department of Surgery, Chang Gung Memorial Hospital, Medical College of Chang Gung University, Taoyuan 333, Taiwan; chensc@cgmh.org.tw; 3Department of Pathology, Chang Gung Memorial Hospital, Medical College of Chang Gung University, Taoyuan 333, Taiwan; susie.ueng@gmail.com

**Keywords:** breast microcalcifications, ductal carcinoma in situ, underestimation, vacuum-assisted breast biopsy

## Abstract

The mammographic appearance of ductal carcinoma in situ (DCIS) is mostly observed as microcalcifications. Although stereotactic vacuum-assisted breast biopsy (VABB) is a reliable alternative to surgical biopsy for suspicious microcalcifications, underestimation of VABB-proven DCIS is inevitable in clinical practice. We therefore retrospectively analyzed the variables in the prediction of DCIS underestimation manifesting as microcalcifications only proved by stereotactic VABB. In 1147 consecutive VABB on microcalcification-only lesions from 2010 to 2016, patients diagnosed with DCIS were selected to evaluate the underestimation rate. The analyzed variables included clinical characteristics, mammographic features, VABB procedure, and biomarkers. Univariate and multivariate analyses were used, and a *p* value < 0.05 was considered statistically significant. Of the 131 VABB-proven DCIS, 108 cases were diagnosed with DCIS and 23 were upgraded to invasive ductal carcinoma (IDC) after subsequent surgery. The small extent of microcalcification, grouped microcalcifications distribution, nearly complete microcalcification removal, and non-calcified specimens without DCIS were low for DCIS underestimation. Among them, the results of non-calcified specimens with or without DICS were the only statistically significant variables by multivariate logistic regression. These results indicate that the histology of non-calcified specimens was highly predictive of DCIS underestimation. Specimens without DCIS had a low upgrade rate to IDC.

## 1. Introduction

Stereotactic vacuum-assisted breast biopsy (VABB) is the standard for the diagnosis of suspicious malignant breast microcalcifications. Meanwhile, there are no advanced imaging diagnostic modalities that can replace the histological diagnosis. This mammography-guided biopsy can be used to diagnose asymptomatic noninvasive or invasive cancers manifesting only with microcalcifications that have been proven to efficiently reduce the mortality of breast cancer [1].

Minimally invasive percutaneous core needle biopsy is a cost-effective and reliable alternative to surgical biopsy for tissue sampling of suspicious breast lesions, regardless of the screening or clinical diagnostic context [2]. Although VABB is a promising technique for obtaining abundant tissues for microscopic evaluation [3,4], the underestimation of atypia lesions (including atypical ductal hyperplasia and flat epithelial atypia) to ductal carcinoma in situ (DCIS) has been reported to range from 0% to 21% for flat epithelial atypia [5,6,7] and 10% to 29% for atypical ductal hyperplasia [8,9,10]. To avoid delayed treatment of early cancers, management with subsequent surgical biopsy should be recommended as guidance [11,12,13], and this has been universally accepted.

In other aspects of clinical consideration, this study aimed to investigate another issue of DCIS underestimation. A meta-analysis of 7350 cases of DCIS including masses or microcalcifications from 52 studies reported a 30.3% underestimation rate of invasive carcinoma for 14-gauge core needle biopsy and 18.9% for 11-gauge VABB [14]. Although the priority of surgical treatment is universal for noninvasive or invasive breast cancer, the performance of sentinel lymph node biopsy is different between DCIS and invasive ductal cancers (IDCs). According to the guidelines on sentinel lymph node performance, sentinel lymph node biopsy should be essential for IDC, but not for DCIS [15]. To provide an assessment to predict the underestimation of biopsied DCIS, we retrospectively reviewed the results of VABB in cases with breast microcalcifications only and analyzed the variables from patient characteristics, mammographic features, VABB procedural relationships, and biopsy biomarkers among the VABB-proven DCIS. Knowingly, the assessment of specimens without calcification was first evaluated.

## 2. Materials and Methods

### 2.1. Patients

This retrospective study was approved by the Institutional Review Board of our hospital (Chang Gung Memorial Hospital, Linkou Medical Center). All cases were clinically based, and the requirement for informed consent was waived. From the data bank of stereotactic VABB on breast microcalcifications between January 2010 and December 2016, 1147 consecutive cases were reviewed. To analyze the parameters in predicting the underestimation of biopsied DCIS, the following inclusion criteria were used: (1) All cases had been diagnosed with DCIS by stereotactic VABB and subsequently received surgery within 2 weeks; (2) all the biopsied targets manifested as microcalcifications only (without associated masses) on mammograms; (3) none of the patients had any palpable mass or associated sonographic mass after evaluation by breast surgeons; (4) all biopsies were performed with VABB, either with 7-gauge or 10-gauge biopsy needles; (5) all procedures had documented successful calcification retrieval by specimen mammograms; and (6) all cases had individual diagnoses on the specimens with and without microcalcifications.

### 2.2. Stereotactic Breast Biopsy on Microcalcifications

All patients signed an agreement for stereotactic guided breast biopsy according to the regulations of our hospital. Stereotactic biopsies were performed by radiologists with at least 10 years of experience in mammographic interpretation and stereotactic guided core needle breast biopsy using digital mammographic devices (Lorad, Selenia, Bedford, MA, USA) with an add-on stereotactic biopsy unit (Lorad, Danbury, NY, USA). With the paired mammographic projections at +15° and −15°, the coordinates (x, y, and z axis) of the selected target could be obtained by computed calculation. Under local anesthesia with 2% lidocaine (5–10 mL) injection over puncture sites, the biopsies were performed with vacuum-assisted biopsy devices (Vacora or Encor; Bard, Irvine, CA, USA). Before termination of the procedure, radiography of the obtained specimens was routinely performed using a mammographic machine to confirm the retrieval of microcalcifications. Matching the specimen radiograms, specimens with (calcified) and without (non-calcified) microcalcifications were separately picked into two different formalin-fixed bottles and then individually sent for microscopic evaluation. Pathologies were separately reported according to calcified specimens and non-calcified specimens.

### 2.3. Subsequent Surgical Excision

Surgical excision was the first recommended procedure for all biopsy-proven malignancies. In operating the DCIS, a wide excision or partial mastectomy was used to document the complete removal of residual cancer cells. The excised specimen was routinely marked with silk stitches or oriented at the medial, lateral, superior, and inferior boundaries, so that the residual microcalcifications or cancer cells could be localized. After taking the specimen radiography, re-excision was immediately followed in cases with microcalcifications near the edge of the specimen. Another means to confirm the complete removal of residual cancer was the microscopic free-cancer margin. A breast pathological specialist reviewed all the involved cases to confirm the diagnoses of DCIS or IDC.

### 2.4. Data Analysis

Underestimation of DCIS was determined in patients with VABB-proven DCIS who were pathologically upgraded to IDC after subsequent surgery. The flowchart of the study is shown in Figure 1.

The variables for statistical analysis included clinical characteristics (age, side of the breast, breast cancer history), mammographic features (breast density, microcalcification extent, morphology and distribution of microcalcifications, diagnosis categories), procedural relationships (VABB needle size, percentage of calcification retrieval, diagnoses of non-calcified specimens), and histopathology (DCIS grades, status of estrogenic and progesterone receptors or HER-2).

The results of the variables were counted from the internal web of medical records. The mammographic features were standardized according to the Breast Imaging Reporting and Data System (BI-RADS) established by the American College of Radiology [16]. The breast density was divided into (1) mostly entirely fatty, (2) homogenous fibroglandular density, (3) heterogeneous fibroglandular dense breast, and (4) extremely dense breast. The microcalcifications extension indicated the longest distance either on the craniocaudal or mediolateral oblique view of the mammogram. The morphologies of microcalcifications were classified as amorphous, pleomorphic, and fine linear or branched. When the microcalcifications are polymorphous, the morphology of microcalcifications with a higher cancer probability would be recorded, in the sequence of increasing cancer probability from amorphous, pleomorphic to linear/ductal. The distributions of microcalcifications included group, regional, and liner or segmental patterns. The final diagnosis categories consisted of BI-RADS 4a, 4b, or 4c.

The gauges of the VABB needles used were either 10-G or 7-G. Microcalcification retrieval was expressed in percentages of ≥90% and <90%, by comparing the targeted microcalcifications on mammography before and after biopsy. Regarding pathologic results, all DCIS could be diagnosed by calcified specimens. Overall, not all non-calcified specimens could be pathologically diagnosed as DCIS. Therefore, we only analyzed the non-calcified specimens.

### 2.5. Statistical Analysis

The association between documented variables among DCIS and upgraded invasive ductal carcinoma was compared using Pearson’s chi-squared test. The cutoff length for microcalcifications extent was selected to yield the highest possible Youden Index score (sensitivity + specificity − 1). The area under the receiver operating characteristic (ROC) curve for microcalcifications extent was calculated. A logistic regression model was used for multivariate analysis. *p* Values ≤ 0.05 were considered statistically significant. All statistical analyses were performed using SPSS software, version 20.0 (SPSS Inc., Chicago, IL, USA).

## 3. Results

The variables from the clinical characteristics, mammographic features, procedural relationships, and tissue biomarkers are listed in Table 1.

### 3.1. Clinical Characteristics

Of the 145 VABB cases, 131 patients received subsequent operation in our hospital, and the other 14 patients did not (Figure 1). The final surgicopathology documented 108 pure DCIS and 23 invasive cancers. All patients were female, and the median age of the patients was 52 years. Stereotactic VABB was performed in 50 (51%) right breasts and 48 (49%) left breasts. Three patients (3.1%) had a history of contralateral breast cancer, and five (5.1%) had a first-degree family history of breast cancer.

### 3.2. Mammographic Features

The breasts were predominantly classified into dense categories (80.1%) and non-dense categories (19.9%). The median length of microcalcifications was 12 mm (range, 5–66 mm). Microcalcifications appeared as amorphous in 50 (38.2%), pleomorphic in 69 (52.7%), and fine linear or branched in 12 (9.2%), and the distributions were group in 96 (73.3%), regional in 18 (13.7%), and liner or segmental in 17 (13%). Seventy-three (55.7%) cases were classified as BI-RASDS 4a, 31 (23.7%) cases to 4b, and 27 (20.6%) cases to 4c.

### 3.3. Procedural Relationships

We undertook the VABB with 7-gauge biopsy needle in 27 and 10-gauge needle in 104 patients. Among them, microcalcification retrieval was achieved in ≥90% in 50 (38.2%) and <90% in 81 (61.8%). All calcified specimens could be diagnosed as DCIS; however, only 66 (50.4%) non-calcified specimens revealed DCIS, and 65 (49.6%) did not.

### 3.4. Pathological Findings

The DCIS grades were documented to be low in 18 (13.7%), intermediate in 81 (61.8%), and high in 32 (24.4%). The estrogen receptors were positive in 103 (78.6%) patients, while the progesterone receptors were positing in 96 (73.3%) patients, and HERS-2 in 38 (29%) patients.

### 3.5. Univariates among DCIS and Upgraded Invasive Carcinoma

The differences in variables among DCIS and invasive ductal carcinomas are listed in Table 2. The extent of microcalcifications, distribution of microcalcifications, percentage of microcalcification retrieval, and the histology of non-calcified specimens, estrogen receptor, progesterone receptor, and HER-2 were statistically different between DCIS and upgraded IDC.

The optimal cut off of microcalcifications extent was 11.5 mm. The DCIS upgraded to IDC was 6.5% when the microcalcifications extent was <11.5 mm, which was statistically significant, to 27.5% with an extent ≥11.5 mm. Among the distributions of microcalcifications, the group pattern of microcalcifications had the lowest upgrade percentage (12.5%) as compared to regional or linear or segmental. When the microcalcification retrieval was ≥90%, the upgrade rate decreased to 6% from 24.7% to >90%. For the non-calcified specimens, whether they contained DCIS or not, the upgrade rates to IDC were 31.8% for those with DCIS and 3.1% for those without DCIS. The negative estrogen receptor, negative progesterone receptor, and positive Her-2 had higher upgrade rates.

Using multivariate logistic regression, only the non-calcified specimen was statistically significant for predicting the upgrade of DCIS to invasive carcinoma. The odds ratio was 1 to 21.492 (95% CI 0.969–116.649, <0.001) (Table 3).

## 4. Discussion

DCIS, also known as intraductal carcinoma, indicates the presence of abnormal cells confined within the milk duct, which is distinguishable from invasive ductal carcinoma. Most of them are asymptomatic or without palpable masses, which often only present as microcalcifications on mammograms. Although such in situ carcinoma refers to the “preinvasive carcinoma” status, excision needs to be performed in which 20% to 30% of those who do not receive treatment developed invasive carcinoma [17]. The operative methods, either with conservative or total mastectomy, are the same in DCIS or IDC; however, knowledge of predicting the underestimation of VABB proved that DCIS facilitates preoperative planning. Basically, it is not advised to undergo sentinel lymph node biopsy with pure DCIS. Obviation of the supplementary performance of sentinel lymph node biopsy will provide benefits including shortening of the operative time, avoiding unnecessary exposure to radiation dose, or minimizing the potential complications of lymph node resection.

In our series, 17.55% of 131 VABB-diagnosed DCIS were microscopically revealed to be invasive components after subsequent surgery. Preoperative mammograms were used for the review. About 80% of breasts were classified as dense and 20% as non-dense. The extent and distribution of microcalcifications were statistically significant between pure DCIS and upgraded IDC. The optimal cutoff size was found to be 11.5 mm. When the extent was smaller than 11.5 mm, the DCIS underestimation was significantly low (6.5%). Nevertheless, other studies have suggested that 30 mm or 40 mm might have a higher probability of underestimation [18,19]. It is understood that a trend for increased size was suspected among the underestimated cases, but this is still controversial, as statistical significance and insignificance have never been reported [20,21]. The cut off DCIS underestimation was hard to define, and should preferably be dependent on the sample collection.

Microcalcifications in group pattern (defined as more than five microcalcifications gathered within a 2 cm^2^ area on the mammogram) were the most widely classified in our series, accounting for 73.3%. Among the distribution patterns, the regional pattern had the highest upgrade percentage to IDC (38.9%), followed by the linear or segmental pattern (23.5%) and group pattern (12.5%). This could be explained by larger territories having a higher chance of underestimation.

The tissue amount obtained by biopsy procedures is an important factor for correct microscopic diagnosis. Because of the advanced development of spring-loaded to vacuum-assisted needles, it has become easier to obtain larger tissues for examination. Complete or mostly complete removal of targeted microcalcifications has become common. Achievement depends on the extent of microcalcification. However, complete removal of targeted microcalcifications has not been standardized for procedure termination. In contrast, biopsy-induced bleeding is practically considered, and post-biopsy hematoma often obscures any residual microcalcifications on the postprocedural mammogram. Thus, we simply assessed nearly (≥90%) and below nearly (<90%) complete microcalcification removal. Our results support the nearly complete removal of microcalcifications, as this had a lower underestimation rate than the below nearly complete removal (6% versus 24.7%). Of course, nearly complete microcalcification removal was exclusive to a small extent of grouped microcalcifications. In certain cases of a larger size, an increased number of core specimens would obtain more microcalcifications, which might lower the underestimation rate [22,23]. Unfortunately, the number of core specimens could not guarantee the percentage of microcalcification retrieval, and the number of samples was dependent on the operator’s decision.

There were no statistically significant differences in DCIS underestimation between the sizes of biopsy needles, which included 7-gauge and 10-gauge needles in our series. However, specimens with or without microcalcification would almost certainly be obtained regardless of the biopsy needle used, 7-gauge or 10-gauge, and they would receive their individual diagnoses [24]. The calcified specimens were more valuable to cancer diagnosis than non-calcified specimens [25,26]. Cheung et al. reported remarkable differences in diagnostic accuracies (91.54% versus 69.49%) after comparing calcified specimens to non-calcified specimens in individual cases [25]. This is in agreement with Margolin et al., who demonstrated that cores with calcification on specimen radiographs were more likely to enable a final diagnosis of malignancy than were cores without calcification (84% versus 71%) [26]. In this analysis, we found that the non-calcified specimens were predictive of DCIS underestimation. There was only a 3.1% upgrade rate for non-calcified specimens diagnosed as non-cancerous diagnoses, and a 31.8% upgrade rate in those with DCIS diagnoses (*p* value < 0.0001). This result was also supported by multivariate logistic regression. To our knowledge, such results have not been published previously. We explained the abnormal cancer cells confined within the ductules of the breast that were less extensive to the neighboring breast tissues as compared to IDC. Conversely, the non-calcified specimen revealed with DCIS indicates larger cancer involvement near the sites of biopsy.

## 5. Conclusions

In cases of DCIS manifesting with microcalcifications only on mammograms, the small extent of microcalcifications, grouped microcalcifications distribution, nearly complete microcalcification removal, and non-calcified specimens without DCIS had lower DCIS underestimation. The non-calcified specimens were highly predictive of DCIS underestimation. Specimens without DCIS had a low upgrade rate to invasive cancers.

## Figures and Tables

**Figure 1 jcm-09-02999-f001:**
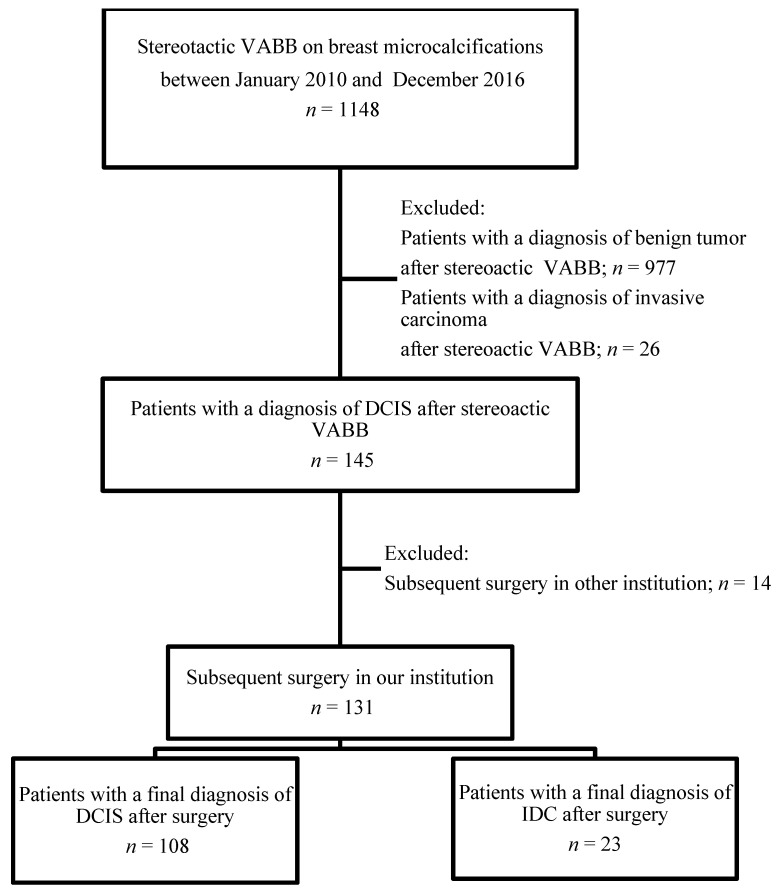
Flow diagram summarizing the total number of vacuum-assisted breast biopsies on suspicious microcalcifications only, and the final diagnosis after surgery. VABB: vacuum-assisted breast biopsy; DCIS: ductal carcinoma in situ; IDC: invasive ductal carcinoma.

**Table 1 jcm-09-02999-t001:** Variables of clinical characters, mammographic features, procedural relationships and specimen biomarkers.

Variables		No. (%)
Study Period	2010–2016	131 (100)
Age of initial diagnosis (years)	Median (IQR)	52.0 (10.0)
Lesion location	Right	70 (53.4)
	Left	61 (46.6)
Family history of breast cancer	Yes	10 (7.6)
	No	121 (92.4)
Parenchymal density	Almost entirely fat	2 (1.5)
	Scattered fibroglandular	24 (18.3)
	Heterogeneously dense	84 (64.1)
	Extremely dense	21 (16.0)
Microlcalcifications extent (mm)	Median (IQR)	12.0 (13.0)
Distribution of microcalcifications	Regional	18 (13.7)
	Grouped	96 (73.3)
	Linear or segmental	17 (13.0)
Morphology of microcalcifications	Amorphous	50 (38.2)
	Pleomorphic	69 (52.7)
	Fine linear or branched	12 (9.2)
Biopsy needle gauge	7 G	27 (20.6)
	10 G	104 (79.4)
Microcalcification retrieval (%)	<90	81 (61.8)
	≥90	50 (38.2)
BI-RADS category	4a	73 (55.7)
	4b	31 (23.7)
	4c	27 (20.6)
DCIS grade	Low	18 (13.7)
	Intermediate	81 (61.8)
	High	32 (24.4)
Histology of non-calcified specimens	Benign	65 (49.6)
	DCIS	66 (50.4)
Estrogen receptor	Negative	28 (21.4)
	Positive	103 (78.6)
Progesterone receptor	Negative	35 (26.7)
	Positive	96 (73.3)
HER-2 status	Negative	93 (71.0)
	Positive	38 (29.0)

Abbreviations: IQR: interquartile range; DCIS: ductal carcinoma in situ; BI-RADS: Breast Imaging Reporting and Data System; HER-2: human epidermal growth factor receptor 2.

**Table 2 jcm-09-02999-t002:** Univariate analysis of the DCIS and upgraded IDC.

Variables	DCIS (*n* = 108)	IDC (*n* = 23)	*p* Value
Age			0.142
≤50	46 (88.5)	6 (11.5)	
>50	62 (78.5)	17 (21.5)	
Lesion location			0.744
Right	57 (81.4)	13 (18.6)	
Left	51 (83.6)	10 (16.4)	
Family history of breast cancer			0.208
Yes	10 (100.0)	0	
No	98 (81.0)	23 (19.0)	
Parenchymal density			0.420
Almost entirely fat	1 (50.0)	1 (50.0)	
Scattered fibroglandular	19 (79.2)	5 (20.8)	
Heterogeneously dense	72 (85.7)	12 (14.3)	
Extremely dense	16 (76.2)	5 (23.8)	
Microcalcifications extent (mm)			0.002
<11.5	58 (93.5)	4 (6.5)	
≥11.5	50 (72.5)	19 (27.5)	
Distribution of microcalcifications			0.021
Regional	11 (61.1)	7 (38.9)	
Grouped	84 (87.5)	12 (12.5)	
Linear or segmental	13 (76.5)	4 (23.5)	
Morphology of microcalcifications			0.195
Amorphous	45 (90.0)	5 (10.0)	
Pleomorphic	54 (78.3)	15 (21.7)	
Fine linear or branched	9 (75.0)	3 (25.0)	
Biopsy needle gauge			0.783
7 G	23 (85.2)	4 (14.8)	
10 G	85 (81.7)	19 (18.3)	
Microcalcification retrieval (%)			0.006
<90	61 (75.3)	20 (24.7)	
≥90	47 (94.0)	3 (6.0)	
BI-RADS category			0.345
4a	63 (86.3)	10 (13.7)	
4b	25 (80.6)	6 (19.4)	
4c	20 (74.1)	7 (25.9)	
DCIS grade			0.221
Low	17 (94.4)	1 (5.6)	
Intermediate	67 (82.7)	14 (17.3)	
High	24 (75.0)	8 (25.0)	
Histology of non-calcified specimens			<0.0001
Benign	63 (96.9)	2 (3.1)	
DCIS	45 (68.2)	21 (31.8)	
Estrogen receptor			0.009
Negative	18 (64.3)	10 (35.7)	
Positive	90 (87.4)	13 (12.6)	
Progesterone receptor			0.045
Negative	25 (71.4)	10 (28.6)	
Positive	83 (86.5)	13 (13.5)	
HER-2 status			0.001
Negative	83 (89.2)	10 (10.8)	
Positive	25 (65.8)	13 (34.2)	

Abbreviations: DCIS: ductal carcinoma in situ; IDC: invasive ductal carcinoma; BI-RADS: Breast Imaging Reporting and Data System; HER-2: human epidermal growth factor receptor 2.

**Table 3 jcm-09-02999-t003:** Multivariate logistic regression in predicting DCIS underestimation.

Variables	Multivariate		
	Odds Ratio	95% CI	*p* Value
Microcalcifications extent (mm)			
<11.5	1		
≥11.5	1.734	0.290–10.373	0.546
Distribution of microcalcification			
Grouped	1		
Regional	1.788	0.365–8.775	0.474
Linear or segmental	0.560	0.098–3.215	0.515
Microcalcification retrieval (%)			
<90	3.452	0.577–20.637	0.175
≥90	1		
Histology of non-calcified specimens			
Benign	1		
DCIS	21.492	3.960–116.649	<0.001
Estrogen receptor			
Negative	10.267	0.520–202.855	0.126
Positive	1		
Progesterone receptor			
Negative	0.117	0.006–2.408	0.165
Positive	1		
HER-2 status			
Negative	1		
Positive	2.606	0.649–10.466	0.177

Abbreviations: DCIS: ductal carcinoma in situ; CI: confidence interval; HER-2: human epidermal growth factor receptor 2.

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
