# Peer review of "Ductal Carcinoma In Situ Underestimation of Microcalcifications Only by Stereotactic Vacuum-Assisted Breast Biopsy: A New Predictor of Specimens without Microcalcifications"

_jcm, 2020, doi:10.3390/jcm9092999_

Round 1

Reviewer 1 Report

The manuscript “Ductal carcinoma in situ underestimation of microcalcifications only by stereotactic vacuum-assisted breast biopsy: A new predictor of specimens without microcalcifications” by Cheung et al. describes the underestimation of ductal carcinoma in situ in non-calcified specimens. The topic of the manuscript is of the potential interest for the readers of Journal of Clinical Medicine. I provided a minor suggestion for the authors to re-write the abstract in order to make the objective of the study more clear for the readers.

Author Response

Reviewer 1 comment:

The manuscript “Ductal carcinoma in situ underestimation of microcalcifications only by stereotactic vacuum-assisted breast biopsy: A new predictor of specimens without microcalcifications” by Cheung et al. describes the underestimation of ductal carcinoma in situ in non-calcified specimens. The topic of the manuscript is of the potential interest for the readers of Journal of Clinical Medicine. I provided a minor suggestion for the authors to re-write the abstract in order to make the objective of the study more clear for the readers.

Reply: I agree with reviewer’s opinion that this manuscript is potential interest to the readers of JCM. I like to rewrite the abstract to more clarify the aim of this study.

Thank you for your comment.

Reviewer 2 Report

This is a retrospective study that involves 145 patients with ductal carcinoma in situ (DCIS) and diagnosed by stereotactic mammography-guided vacuum-assisted breast biopsy (VABB). The aim was to provide an assessment to predict the underestimation of biopsied DCIS. Using multivariate logistic regression, the authors stated that only the non-calcified specimen was statistically significant for predicting the upgrade of DCIS to invasive carcinoma. I have only some minor comments:

  • Line 230. ...was influenced by the scene of the procedure... Please clarify this sentence.
  • Line 232. However, specimens.... Please clarify this sentence
  • I suggest briefly discussing ref 23 and 24. 
  • Line 179. The non-calcified specimen with diagnosis of DCIS... Please clarify this sentence.

Author Response

Reviewer 2 comment:

This is a retrospective study that involves 145 patients with ductal carcinoma in situ (DCIS) and diagnosed by stereotactic mammography-guided vacuum-assisted breast biopsy (VABB). The aim was to provide an assessment to predict the underestimation of biopsied DCIS. Using multivariate logistic regression, the authors stated that only the non-calcified specimen was statistically significant for predicting the upgrade of DCIS to invasive carcinoma. I have only some minor comments:

Thank you very much for comments. Please find replies as below:

Line 230. ...was influenced by the scene of the procedure... Please clarify this sentence.

Reply: Sorry for the redundant sentence. We like to delete the ‘scene of the procedure’, The number of samples were depended on the operator’s decision that had been modified on revised manuscript.

Line 232. However, specimens.... Please clarify this sentence

I suggest briefly discussing ref 23 and 24.

Reply: We added a reference 23 to the revised manuscript. On the revised manuscript, we add relevant data from references 23 and 24 (24 and 25 on revised manuscript). “Cheung et al, reported remarkable differences in diagnostic accuracies (91.54% versus 69.49%) after compared calcified specimens to non-calcified specimens in individual cases [24]. Sympathized with Margolin et al. who documented cores with calcification on specimen radiographs were more likely to enable a final diagnosis of malignancy than were cores without calcification (84% versus 71%) [25].

Line 179. The non-calcified specimen with diagnosis of DCIS... Please clarify this sentence.

Reply: Sorry for unclear sentence, we had modified this sentence to “For the noncalcified specimens whether containing DCIS or not, the upgrade rates to IDC were 31.8% for those with DCIS and 3.1 % for those without DCIS.’